# Glycyrrhetinic Acid Improves Insulin-Response Pathway by Regulating the Balance between the Ras/MAPK and PI3K/Akt Pathways

**DOI:** 10.3390/nu11030604

**Published:** 2019-03-12

**Authors:** Yuan Zhang, Shengnan Yang, Man Zhang, Zhihua Wang, Xin He, Yuanyuan Hou, Gang Bai

**Affiliations:** State Key Laboratory of Medicinal Chemical Biology, College of Pharmacy and Tianjin Key Laboratory of Molecular Drug Research, Nankai University, Haihe Education Park, 38 Tongyan Road, Tianjin 300353, China; yuanyuannankai621@163.com (Y.Z.); m18822186169@163.com (S.Y.); 18202572363@163.com (M.Z.); 15822278821@163.com (Z.W.); xinhe@mail.nankai.edu.cn (X.H.); houyy@nankai.edu.cn (Y.H.)

**Keywords:** glycyrrhetinic acid, insulin resistance, cross-talk, Ras/MAPK, PI3K/AKT

## Abstract

Glycyrrhetinic acid (GA), a bioactive component in the human diet, has been reported to improve hyperglycemia, dyslipidemia, insulin resistance and obesity in rats with metabolic syndrome. However, GA-specific target proteins and the mechanisms involved in the downstream signaling and cross-talk to improve insulin sensitivity have not been fully elucidated. In this study, the potential targets of GA were identified by chemical proteomics strategies using serial GA probes for target fishing and cell molecular imaging. Intracellular enzyme activity evaluation and insulin resistance models were used for validating the function of the target proteins on the downstream insulin signaling pathways. Collectively, our data demonstrate that GA improved the insulin-responsive pathway and glucose consumption levels via multiple diabetogenic factors that activated the insulin signaling pathway in HepG2 cells. GA improved Glucose transporter 4(GLUT4) expression by targeting the Ras protein to regulate the mitogen-activated protein kinase (MAPK) pathway. GA exhibited a strong inhibitory effect on IRS1ser307 phosphorylation in cells treated with the Protein kinase C (PKC) activator Phorbol 12-myristate 13-acetate (PMA.) Consistently, IRS1ser307 phosphorylation was also inhibited by GA in Free fatty acid (FFA)-treated HepG2 cells. GA also inhibited the PMA-induced phosphorylation of IκB kinase α/β (IKKα/β), c-Jun N-terminal kinase (JNK) and p38 proteins (P38), suggesting that IKKα/β, JNK and P38 activation is dependent on PKC activity.

## 1. Introduction

Insulin resistance causes target tissues to fail to respond properly to normal levels of circulating insulin, which is a common pathological condition known as type 2 diabetes mellitus (T2DM). When insulin binds to its cognate receptor (transmembrane tyrosine kinase receptor), it leads to the autophosphorylation and activation of the insulin receptor (IR). The activated IR phosphorylates several insulin receptor substrate (IRS) protein families. The IRS proteins (IRS-1 to IRS-4) play a crucial role in insulin signaling [1]. For example, IRS-1 function is multifaceted and involves the phosphorylation of IRS-1 at multiple serine/threonine residues [2]. Recent studies show that diabetogenic factors, such as free fatty acid (FFA) [3], TNF-α [4] and hyperinsulinemia, increase the serine phosphorylation of IRS-1, and ser307/612/632 were identified as phosphorylated sites [2,5]. The phosphorylation of key sites, such as IRS1ser307, represent key targets. The importance of the IRS1ser307 site is that it is involved in insulin resistance in hyperinsulinaemia or acute lipid infusion [6,7]. Therefore, the phosphorylation of IRS-1 at ser307 is considered to be a major player in insulin signaling and insulin resistance. 

Insulin signaling has been studied for many years, and the phosphoinositide 3-kinase (PI3K)/AKT and Ras/MAPK pathways play a major role downstream of the IRS [8]. Activation of the MAPK signaling cascade down-regulates glucose transporter type 4 (GLUT4) expression, resulting in decreased glucose transport. The PI3K/Akt pathway plays a key role in mediating insulin metabolism, especially by increasing glucose uptake through the insulin sensitive GLUT4 [9]. Moreover, the interaction between the Ras/MAPK and PI3K/Akt pathways mediates the metabolism of glucose and lipids and leads to oxidative stress [10]. Abnormalities in both pathways lead to abnormal glucose and lipid metabolism, which is the main pathophysiological feature of T2DM [11].

The roots and rhizomes of licorice (*Glycyrrhiza glabra* Linn.) species have been used extensively as natural sweeteners and herbal medicines. Glycyrrhetinic acid (GA) is a triterpenoid saponin, which is the main bioactive component of licorice root and is known to have anti-inflammatory, antidiabetic and antitumor pharmacological effects [12]. Previous studies have shown that GA offsets the development of visceral obesity and improves dyslipidemia by selectively inducing the expression of tissue lipoprotein lipase (LPL) [13]. GA reverses insulin resistance, probably by reducing hexose-6-phosphatedehydrogenase (H6PDH) and 11β-hydroxysteroiddehydrogena-setype1 (11β-HSD1) and selectively decreasing phosphoenolpyruvate carboxykinase (PEPCK) activity [14]. GA improved LPL expression, lipid deposition and serum lipids in obese rats on a high-fat diet [15]. GA also displays antiadipogenic and pro-lipolytic effects by modulating Akt and hormone-sensitive lipase (HSL) phosphorylation [16]. However, the accurate protein targets and molecular mechanisms by which GA improves insulin resistance and regulates blood glucose and lipid metabolism remain unclear. 

In this study, the potential targets of GA were identified by chemical biology strategies using synthetic GA probes for click reaction, fishing rod and cell molecular imaging. Intracellular enzyme activity evaluations and insulin resistance models validated the function of the potential target proteins on downstream insulin signaling pathways. The results demonstrated that GA affected the action of Ras GTPases and PKC, regulated the cross-talk between the Ras/MAPK and PI3K/Akt signaling pathways, promoted GLUT4 expression, reduced inflammation and improved insulin sensitivity.

## 2. Materials and Methods 

### 2.1. Animals

Six-week-old female Kunming mice, weighing 20–25 g, (specific pathogen-free (SPF)) were purchased from the Laboratory Animal Experimental Center of Academy of Military Medical Sciences (SCXK2012-0004, Beijing, China). Studies using the experimental animals were performed according to protocols approved by Nankai University Ethics Committee on Pre-Clinical Studies. The mice were randomly assigned to 4 groups (*n* = 10 mice per group). Three groups were intragastrically (i.g.) administered GA (high, 100 mg/kg; mid, 50 mg/kg, low, 25 mg/kg) daily for 14 days, and the control group was i.g. administered saline (0.5 mL/day). The mice were fed mouse chow throughout the treatment period of two weeks. The posterior orbital venous plexus approach was applied to collect blood samples for measuring the glucose, glucocorticoid, insulin, IL-6 and TNF-α levels, which were determined using an ELISA kit. The ELISA kits were purchased from assay company (Beijing, China). Membrane and Cytosol Protein Extraction Kit, and BCA Protein Assay Kits were purchased from Beyotime Biotechnology Corporation (Shanghai, China).

### 2.2. Reagents, Cell Culture, SDS-PAGE, and Western Blotting

HepG2 (hepatocellular carcinoma cells) cell lines were obtained from the American Type Culture Collection (Manassas, VA, USA). HepG2 cell lines were maintained in Dulbecco’s Modified Eagle’s medium (DMEM) supplemented with 10% (*v*/*v*) fetal bovine serum and 100 units/mL penicillin and were incubated at 37 °C in a 5% CO_2_ atmosphere. Protein lysates were obtained according to the manufacturer’s suggested protocol of the Membrane and Cytosol Protein Extraction Kit, and the concentration of proteins was detected by a BCA Protein Assay Kit (Solarbio, Beijing, China). The western blot and SDS-PAGE analyses were performed as previously described [17]. Fe_3_O_4_ amino magnetic microspheres (MMs) were purchased from Tianjin baseline Chromtech Research Center (Tianjin, China). The primary antibodies anti-Ras (#3965), anti-P38 (#9212), anti-phospho-P38 (#9211), anti-JNK (#9252S), anti-phospho-JNK (#9251S), anti-GADPH (#2118), and a goat anti-rabbit IgG (#7074) secondary antibody were purchased from Cell Signaling Technology (Beverly, MA, USA). Anti-PKC (ab179521), anti-HSD11B1 (ab39364), anti-AKT (ab39364), anti-phospho-AKT(473) (ab81283), anti-GSK3β (ab93926), anti-phospho-GSK3β (ab131097) anti-IKKα/β (ab178870), anti-phospho-IKKα/β (ab17943), and Alexa Fluor 594-conjugated goat anti-rabbit IgG (ab150084) antibodies were purchased from Abcam (Cambridge, UK). Anti-GLUT4 (bS3680) was purchased from Bioworld (Minnesota, MN, USA). Anti-IRS1 (phospho-ser307) (bs-2736R), anti-IRS1 (bs-0172R) were purchased from Bioss (Beijing, China). The phorbol 12-myristate13-acetate (PMA) was purchased from YEASEN (Shanghai, China). The farnesyltransferase (FTase) inhibitor tipifarnib (Tip) and the PKC inhibitor bisindolylmaleimide I (GF109203X) were purchased from Selleck (Houston, TX, USA). Glycyrrhetinic acid (purity >98.5%, as determined by HPLC) and propargylamine were purchased from J&K Chemical (Beijing, China).

### 2.3. Enrichment of Target Proteins in Cells

HepG2 cells were maintained in 75-cm^2^ culture flasks and were cultured with 10 µM alkynyl-GA probes for 6 h. After three washes with precooled PBS, 500 µL lysis buffer (Solarbio, Beijing, China) was added to the culture, and the cells were incubated on ice for 30 min. Cell lysates were then treated with GA-modified functionalized magnetic microspheres (probe 1) and were incubated with a catalyst (2.0 mM sodium ascorbic acid and 1.0 mM CuSO_4_ in precooled PBS) overnight at 4 °C to capture the targets of the alkynyl-GA probe. Afterwards, probe 1 was separated with magnets and was washed with PBS. Enriched probe 1 was then treated with moderate DL-dithiothreitol (DTT) (100 mM) at 37 °C for 1 h to release the captured protein targets, which were then identified using SDS-PAGE and subsequent western blotting, which were performed in accordance with a previously reported method [17].

### 2.4. Colocalization of Target Proteins and GA

HepG2 cells were seeded in flasks and were treated with 1 µM alkynyl-GA probes for 6 h. The cells were washed and fixed with 4% paraformaldehyde and were then blocked with 10% goat serum. Anti-Ras (1:1000) and anti-PKC (1:500) antibodies were added, and the culture was incubated overnight at 4 °C. Afterwards, Alexa Fluor 594-conjugated secondary antibodies (1:1000) were added to the cells, and they were incubated for 1 h at room temperature. The N3-tag substrate (10 µM) was then added to the cells for the probe 2 click reaction with the aforementioned catalyst system, and the culture was incubated for 1 h at 37 °C. After an adequate number of washes, fluorescence images were obtained with a confocal microscope (Leica TCS SP8, Japan): Alexa Fluor594: Excitation Wavelength: 594 and Emission Wavelength: 617 nm; probe 2: Excitation Wavelength 488 and Emission Wavelength: 520 nm. DAPI (1:1000) was used to stain the nuclei, and the images were captured at Excitation Wavelength: 405 and Emission Wavelength: 430 nm.

### 2.5. Evaluation of RAS Activation

A New East Assay Kit (NewEast Biosciences, Wuhan, China) was used for Ras detection according to the manufacturer’s instructions. Briefly, approximately 80–90% confluent cultured cells (10-cm plate) were stimulated with GA (0.2, 20 or 20 µM) for 6 h. Then, the culture media was aspirated, and 500 µL of the assay/lysis buffer was added to the cells, and it was incubated on ice for 10–20 min. The lysates were clarified by centrifugation for 10 min (12,000·*g*, at 4 °C). The GTPγS- and GDP-bound proteins served as positive and negative controls, respectively. The cell extracts were divided into two tubes, and 20 µL of 0.5 M EDTA was added. Five microliters of GTPγS was added to the positive control, and 5 µL of GDP was added to the negative control. The tubes were incubated at 30 °C for 30 min, and the loading was terminated by placing the tubes on ice and then adding 60 mM MgCl_2_. Then, an antiactive Ras monoclonal antibody (1 µL) was added to each tube. A resuspended bead slurry was quickly added to each tube. Then, the tubes were incubated at 4 °C for 1 h with gentle agitation. The beads were pelleted by centrifugation for 1 min at 5000·*g*. The supernatant was aspirated and discarded, and the beads were washed 3 times with the assay/lysis buffer. The tubes were boiled for 5 min prior to the SDS-PAGE and western blot analyses.

### 2.6. PKC Kinase Activity Assay Kit

A PKC Kinase Activity Assay Kit (ab139437) was used for PKC detection. The manufacturer’s instructions were followed. First, 50 µL of Kinase Activity Assay buffer was added to each well at room temperature for 10 min after aspirating the liquid from all wells. Then, 30 µL of the samples were added, 10 µL of reconstituted ATP was added to each well and it was incubated at 30 °C for up to 30 min. The reaction was stopped by emptying the contents of each well. Subsequently, 40 µL PKC phosphorspecific substrate antibody was added to each well at room temperature for 60 min, and wash buffer was added to each well three times. Then, 40 µL of the diluted anti-rabbit IgG-HRP conjugate was added, it was incubated at room temperature for 30 min, and wash buffer was added to each well three times. Finally, 60 µL TMB substrate was added to each well, and 20 µL stop solution was added to each well. The absorbance increase was measured on a microplate reader at OD = 450 nm.

### 2.7. Statistical Analysis

The results are reported as the means ± standard deviation (S.D.). An unpaired, two-tailed Student’s *t*-test was applied for the statistical comparison of two groups and one-way ANOVA with a Bonferroni correction was used for multiple comparisons. All data were processed using GraphPad Prism statistical software, version 5.01 (GraphPad Software, San Diego, CA, USA).

## 3. Results

### 3.1. GA Affects Insulin-Responsive Pathwayin the Whole Body

Emerging data suggest that GA can counteract the development of T2DM by improving insulin sensitivity [14]. Our results were consistent with the literature as, compared with the control, the administration of GA dose-dependently and significantly activated glucocorticoid in mice (Figure 1a). Lower blood glucose and insulin levels were also seen in all the treated mice when compared to those in the control mice (Figure 1b,c). Moreover, the inflammatory factors IL-6 and TNF-α were significantly suppressed, especially in the high-dose groups (Figure 1d,e). As shown in Figure 1, the insulin-responsive pathway improved and glucose production decreased as GA exerted glucocorticoid-like effects. To explain this phenomenon, we performed the following experiment to explore GA-target proteins in the insulin signaling pathway.

### 3.2. Prediction and Verification of the Targets of GA 

To elucidate targets of GA to improve insulin resistance, we first accomplished reverse docking of GA. The 3D structure of GA was prepared in the sdf format using the ChemBio3D Ultra 13.0 software (PerkinElmerInc., San Diego, CA, USA) and was submitted to the Pharm Mapper server (http://59.78.96.61/pharmmapper/) with the choice of human protein targets only with the maximum generated conformations set to 300. Next, the first 20 candidate targets were selected to analyze the candidate interactions by String 10.0 (http://www.string-db.org/). As shown in Figure 2A, three targets in the insulin-related signaling pathways, namely, HRAS, PRKCA (PKCα) and MAP2K1 (MEK1) (blue circles), were highly recommended. In addition, two targets in the steroid hormone biosynthesis signaling pathways, HSD11B1 and HSD17B1 (red circles), were also predicted. AutoDock 4.2 software (Olson Laboratory, LaJolla, CA, USA) was then used to perform molecular docking to assess the interaction between the antidiabetic targets and GA. GA displayed different scores for the HRAS, PKCα, MEK1, HSD11B1 and HSD17B1 targets, showing −7.47, −8.77, −9.93, −10.31 and −10.12 kcal/mol, respectively (Appendix A).

To identify the predicted targets, a set of chemical probes were synthesized to locate and capture the target proteins listed above. The synthetic routes for Alkynyl-GA, Probe1 and Probe2 are shown in the Appendix A. In Figure 2B, the GA-modified functionalized MMs (probe 1) were used to capture the target proteins in HepG2 cells. The collection efficiency was measured by SDS-PAGE with Coomassie Brilliant Blue staining. The results showed that some proteins were enriched from the lysate of the HepG2 cells by Probe1 (Figure 2C, left panel lane 3), but in the negative control group, the non-GA modified probe-MMs showed only slight protein detection (Figure 2C, left panel lane 2). This result indicates that the fishing rod strategy selectively enriched and released the related targets by Probe1. Next, a western blot further confirmed that RAS, PKC and HSD11B1 were likely targets that were significantly enriched compared to the negative control group (Figure 2C). These results indicate that Ras and PKC might be potential target proteins of GA in insulin-related signaling pathways. Furthermore, the Probe2 was used for the cytochemical colocalization staining of the Ras and PKC proteins in the HepG2 cells. As shown in Figure 2D, the fluorescence by Alkynyl-GA (1 µM) was obvious in the cytoplasm (green), and the distribution of Ras, stained by Alexa Fluor594 secondary antibodies (red), was observed in the cytoplasm and membrane and partially merged with Alkynyl-GA (yellow), which is indicated by the arrows. 

Normally, GTP binding increases the activity of Ras, and the hydrolysis of GTP to GDP renders it inactive. In our study, we employed a Ras Activation Assay Kit based on the use of a configuration-specific anti-Ras-GTPγS monoclonal antibody to measure active Ras-GTP levels and the precipitated active Ras that was pulled down by protein A/G agarose, as detected by the immunoblot analysis using an anti-Ras antibody. As shown in Figure 2E, 20 μM GA significantly inhibited the intermolecular interactions of GTP-bound Ras and the activation on HepG2 cells. This result indicates that GA inhibited the interactions of GTP-bound Ras and effect of the activation of Ras in HepG2 cells. In addition, to determine the inhibitory effect of GA in PKC, the activation of PKC in HepG2 cells was measured. As shown in Figure 2F, stimulation by Phorbol 12-myristate13-acetate (PMA), a specific PKC agonist, increased the activity of PKC, and the effect was significantly decreased after adding 10 μM GA. The above results suggest that GA might act on RAS and PKC targets and affect their downstream functions.

### 3.3. Effect of GA on Glucose Consumption and Insulin-Responsive Pathway

Emerging data suggest that TNF-α, FFA and hyperinsulinemia, which induce insulin resistance, activate er307 phosphorylation on IRS-1 and inhibit its function. Therefore, we used insulin-stimulated HepG2 cellsto activate the insulin signaling pathway. The glucose consumption amount decreased when the insulin signaling pathway was activated. However, GA improved the levels of glucose consumption in a dose-dependent manner, which was similar to the metformin group (Figure 3A). To further elucidate the role of GA at the molecular level, the insulin-stimulated HepG2 cells were treated with 5 and 10 μM GA overnight, and the phosphorylation levels of the downstream pathways were evaluated. As shown in Figure 3B, the insulin-stimulated HepG2 cells showed that IRS1ser307 was elevated, and AKTser473 was decreased in the model group. In contrast, GA suppressed the activation of the phosphorylation of IRS1ser307 and increased the phosphorylation of AKTser473. 

Studies have shown that TNF-α plays an important role in obesity-induced insulin resistance by promoting the serine phosphorylation of IRS-1 [18,19]. TNF-α acts on multiple signaling nodes by direct and indirect mechanisms in the way of insulin action. In Figure 3C, in the TNF-α activated insulin signaling pathway, the glucose consumption showed a downward trend. However, GA improved the levels of glucose consumption similar to that of metformin. To further elucidate the role of GA, the HepG2 cells were stimulated with TNF-α for 24 h and were then treated with 5 and 10 μM GA overnight. As shown in Figure 3D, the phosphorylation of IKKα/β along with IRS1ser307 was elevated in the model group. However, GA suppressed the activation of the phosphorylation of IKKα/β and IRS1ser307. 

Increased FFA is also associated with obesity and type 2 diabetes. Higher FFA levels induce secretion of inflammatory cytokines and compromise insulin sensitivity [20]. As shown in Figure 3E, glucose consumption decreased in the FFA-stimulated group, but GA reversed the levels of glucose consumption in a manner similar to metformin (Figure 3E). To evaluate the molecular mechanism of the GA-regulated IRS1ser307/GSK3β signaling, the phosphorylation of IRS1ser307 and GSK3β was also investigated. As shown in Figure 3F, the FFA group was significantly greater than the control group, and after treatment with GA, the phosphorylation of IRS1ser307 was significantly reduced. In contrast, the phosphorylation of GSK3β in the FFA group was accordingly diminished. However, the results were reversed by GA in a dose-response manner.

### 3.4. Effect of GA on GLUT4 Expression

A previous study proved that 5-fluoro-farnesylthiosalicylic acid, a Ras inhibitor, induced a significant increase in glucose uptake and expression of the mRNA GLUT4 [21]. To clarify whether GA promotes the expression of GLUT4 protein, the HepG2 cells were incubated with 1 μM insulin for 36 h and were then treated with the farnesyltransferase (FTase) inhibitor tipifarnib (Tip, 5 μM) or GA (5, 10 μM) overnight. As shown in Figure 4a, expression of GLUT4 in the cell cytoplasm of the activated insulin signaling pathway group was dramatically reduced, which led to a marked reduction in cell glucose uptake. However, compared to the activated insulin signaling pathway group, the Tip (5 μM) and GA (5, 10 μM) groups showed substantially elevated GLUT4 expression in the cell cytoplasm. Furthermore, as demonstrated by immunofluorescence staining, both GA (10 μM) and Tip (5 μM) significantly improved the GLUT4expression in HepG2 cells (Figure 4b). 

### 3.5. Effect of GA on Anti-Inflammation and Insulin-Responsive Pathway

The role of PKC in the induction of IRS1ser307 phosphorylation was recently reported [22]. The induction of hepatic insulin resistance by FFA coincides with the membrane translocation of PKC, and its link is also confirmed [23]. A PKC inhibitor blocks the FFA-induced phosphorylation of JNK and IKKα/β, suggesting that JNK and IKKα/β activation is dependent on PKC activity [24]. In this study, FFA activated the insulin signaling pathway, leading to increased phosphorylation of IKKα/β, JNK and p38 in HepG2 cells. As shown in Figure 5a, the phosphorylation of JNK, p38 and IKKα/β were markedly decreased compared with the GA (10 μM) treatment. Furthermore, GA exhibited a strong inhibitory effect on IRS1ser307 phosphorylation, which was stimulated by the PKC activator PMA. GA also inhibited the PMA-induced phosphorylation of IKKα/β and JNK. The result indicated that IKK and JNK activation was dependent on PKC activity, which was similar to the PKC inhibitor (Figure 5b). Above evidence suggests that GA-inhibited PKC activity decreased IRS1ser307 phosphorylation to improve the insulin-responsive pathway in HepG2 cells, which should be regulated by the MAPK and NF-κB signaling pathways.

Chronic inflammation is always accompanied by insulin resistance [25]. TNF-α and IL-6 are the most important pro-inflammatory mediators responsible for inducing insulin resistance in adipocytes and peripheral tissues. The inhibition of TNF-α and IL-6 production may be one of the options for preventing the development of insulin resistance and the pathogenesis of T2DM [26]. As shown in Figure 5c,d, the results showed that the levels of TNF-α and IL-6 in the activated insulin signaling pathway were significantly increased. However, after treatment with GA, TNF-α and IL-6 decreased in a dose-dependent manner. The result indicated that GA also improved insulin-responsive pathway by inhibiting inflammatory factors, such as IL-6, which was similar to the positive control cortisol.

## 4. Discussion

Cell-based studies demonstrate that insulin resistance is usually an impairment of insulin signaling through the phosphorylation of IRS on serine and threonine residues [27]. The activation of insulin receptors by their ligands triggers a series of phosphorylation events. Insulin binding to the α-subunit of the receptor generates conformational changes that induce its catalytic activation and the autophosphorylation of several Tyr residues located in the β-subunit cytosolic region, resulting in the recruitment and phosphorylation of receptor substrates, such as IRS and Shc proteins [28,29]. The typical phosphorylation of IRS1ser307 is thought to mediate the insulin signaling pathway. In our study, the phosphorylation of IRS1ser307 was selected as a key objective, and multiple diabetogenic factors, such as FFA, TNF-α, and insulin, were used to stimulate HepG2 cells and activate the insulin signaling pathway. The results revealed that GA inhibited the activation of the phosphorylation of IRS1ser307 and upregulated the expression of AKTser473 and GSK3β in the PI3K/Akt pathway to improve insulin sensitivity under different insulin resistance conditions. At the same time, GA improved the glucose consumption level, which was similar to the positive control drug metformin. 

After insulin receptor activation, one pathway proceeds through the insulin receptor substrates (IRSs) and relies on the activation of PI3K/Akt/GSK3β to mediate most of the metabolic effects of insulin, regulating glucose transport, gluconeogenesis and lipid synthesis [11]. The other receptor tyrosine kinase pathway proceeds by binding tyrosine-phosphorylated IRS or Shc through to the Grb2/Sos, Ras/MAPK pathway to regulate the gene expression, cellular proliferation, differentiation, apoptosis, inflammation and insulin-associated mitogenic effects [30]. The crosstalk between the PI3K⁄Akt and Ras/MAPK pathways is reported in several experimental models [10]. For example, cichoric acid, a caffeic acid derivative found in Echinacea purpurea, promotes glucosamine-mediated glucose uptake and is inhibited by activating the PI3K-Akt pathway and downregulating the Ras/MAPK signaling pathway [31]. The cross-talk between the RAS/MAPK and PI3K/Akt signaling pathways plays an important role not only in reducing inflammation [32] but also in improving insulin resistance.

Studies have shown that the small family of GTPases (Rab, Ras and Rho) are involved in a diverse array of signal transduction pathways related to glucose uptake in insulin-responsive tissues [33]. A previous study found that the Ras inhibitor F-FTS induces insulin sensitivity and glucose uptake [21]. In the current study, the possibility that GA targeted the Ras protein was clarified via chemical proteomic approaches using GA molecular probes, and the effect of GA on the intermolecular interaction of GTP binding to Ras and reducing the activation of Ras in HepG2 cells was verified. In addition, we observed that GA restored the hyperinsulin-induced impairment of GLUT4 translocation. Therefore, we speculated that GA regulated the expression and transcription of GLUT4 by targeting RAS proteins to improve glucose uptake.

FFA increases diacylglycerol levels, leading to PKC activation, which reduces insulin-mediated IRS and Akt phosphorylation and PI3K activation [34]. The IRS/PI3K/Akt/GSK3β signaling pathway is crucial for the regulation of insulin metabolism. As shown by our data, GA reduced the level of IRS1ser307 phosphorylation and significantly promoted the phosphorylation of GSK3β protein. This result demonstrates that GA reversed insulin resistance by activating the PI3K/Akt pathway. In addition, PKC, IKKα/β and JNK are implicated in phosphorylating IRS proteins and inducing insulin resistance. For members of the PKC family, activated by PMA [35], their activity is mediated by members of the MAPK pathway [36]. FFA, a ligand of the toll-like receptor, activates the PKC-mediated MAPK pathway [37]. Increased FFA levels also increase the activity of IKK and JNK, cause the serine phosphorylation of IRS1ser307 and block IRS tyrosine phosphorylation [38]. Activated PKC also enhances the production of pro-inflammatory mediators, thus increasing the inflammation contributing to obesity-induced insulin resistance [39]. Our results show that GA exhibited a strong inhibitory effect on IRS1ser307 phosphorylation in HepG2 cells, which was stimulated by the PKC specific activator PMA, and induced phosphorylation of IKKα/β and JNK. In addition, GA dose-dependently inhibited the over production of TNF-α and IL-6 in the activated insulin signaling pathway. These results suggest that GA targeting of the PKC protein improved the insulin-responsive pathway by suppressing the MAPK and NF-κB signaling pathways.

In summary, this research demonstrates that GA acted on Ras and PKC targets, regulated GLUT4 expression, glycogen synthesis and alleviated inflammation by the activation of the PI3K/Akt signaling pathway and the suppression of Ras/MAPK, altering the fine balance to improve the insulin-response pathway. Although natural products have many effects on the prevention and treatment of diabetes, the research on their hypoglycemic mechanism needs further research. Meanwhile, side effects and toxicity are still a problem to be considered.

## Figures and Tables

**Figure 1 nutrients-11-00604-f001:**
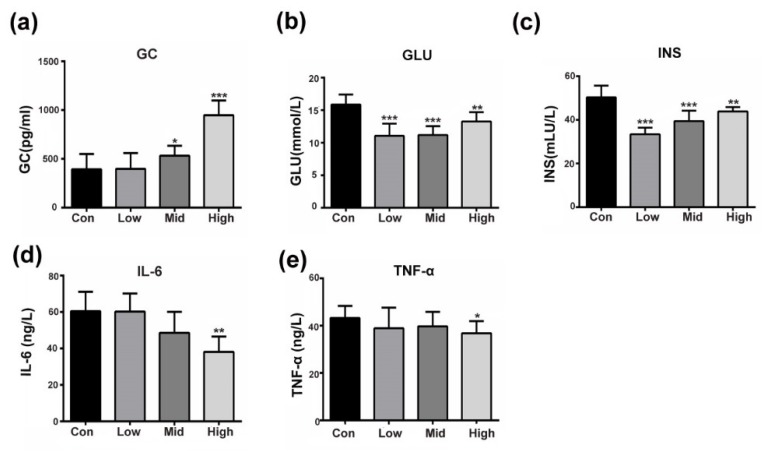
GA affects insulin-responsive pathway in the whole body. (**a**–**e**) Fasting plasma glucocorticoid (GC), insulin (INS), glucose (GLU), IL-6 and TNF-α concentration after GA administration. The mice were treated with saline 0.5 mL/day (control) or GA (High, 100 mg/kg; Mid, 50 mg/kg, Low, 25 mg/kg). The assay was performed as the mean ± S.D. (*n* = 10). * *p* < 0.05, ** *p* < 0.01, *** *p* < 0.001 compared to the control.

**Figure 2 nutrients-11-00604-f002:**
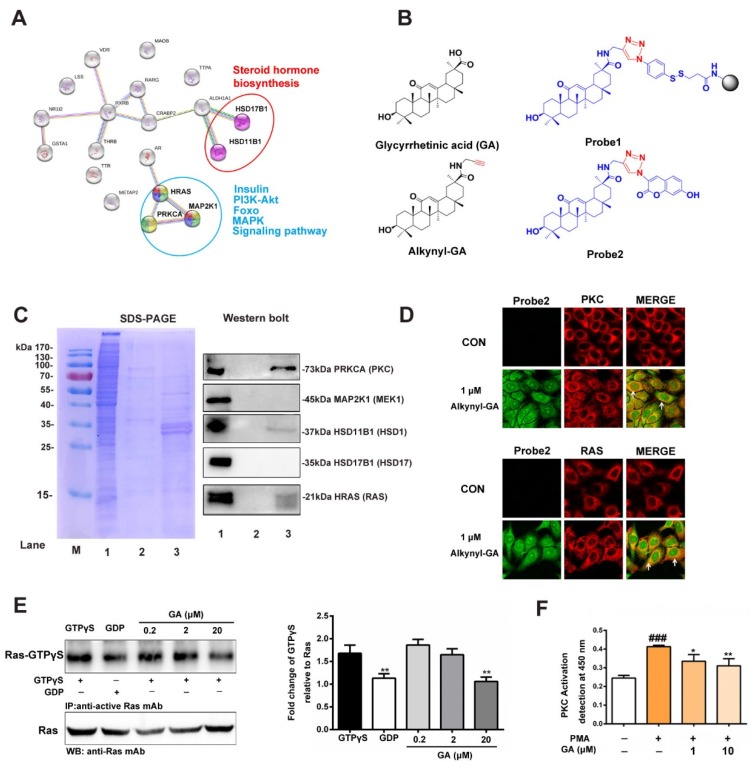
Prediction and verification of the targets of GA. (**A**) The interaction and function assays for GA with the target proteins that were predicted by Pharm Mapper and String 10.0. (**B**) Synthesis of alkynyl-modified GA (Alkynyl-GA), functionalized magnetic microspheres (Probe1) and fluorescent click product (Probe2). (**C**) SDS-PAGE (left panel) and western blotting (right panel) were used to detect the capture capacity of Probe1 for the target protein enrichment. Lane 1 includes the HepG2 lysates as a loading control; Lane 2, fishing lysates by the non-GA probe modified microspheres as a negative control; Lane 3, captured and released proteins by Probe1. All protein samples were of equal cell lysate concentrations before fishing, and the same samples were used for western blotting. (**D**) Colocalization analysis of Alkynyl-GA and RAS and PKC protein by confocal fluorescence microscopy. (**E**) A pull-down system was used to confirm GA that inhibited the binding of GTP-bound in HepG2 cells at 0.2, 2 and 20 μM. (**F**) Effect of GA on PKC activity in HepG2 cells. The assay was performed as the mean ± S.D. (*n* = 3), (* *p* < 0.05, ** *p* < 0.01, ^###^*p* < 0.001).

**Figure 3 nutrients-11-00604-f003:**
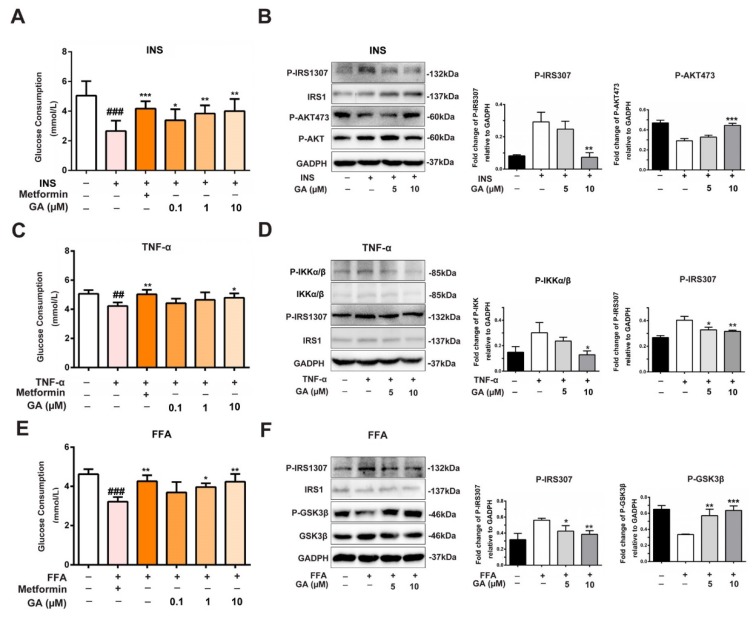
Effect of GA on glucose consumption and insulin-responsive pathway. (**A**) The effect of GA on glucose consumption in the activated insulin signaling pathway. (**B**) A protein level analysis of the upregulation/downregulation of the AKTser473 and IRS1ser307 kinases in the insulin-stimulated HepG2 cells. (**C**) The effect of GA on glucose consumption in the TNF-α-stimulated HepG2 cells. (**D**) A protein level analysis of the upregulation/downregulation of IKK and IRS1ser307kinases in TNF-α-stimulated HepG2 cells. (**E**) The effect of GA on glucose consumption in FFA-stimulated HepG2 cells. (**F**) A protein level analysis of the upregulation/downregulation of the GSK3β and IRS1ser307 kinases in FFA-stimulated HepG2 cells. The error bars indicate the mean ± S.D. (*n* = 3) (* *p* < 0.05, ** *p* < 0.01, ^##^
*p* < 0.01, *** *p* < 0.001, ^###^
*p* < 0.001 compared to the activated insulin signaling pathway group).

**Figure 4 nutrients-11-00604-f004:**
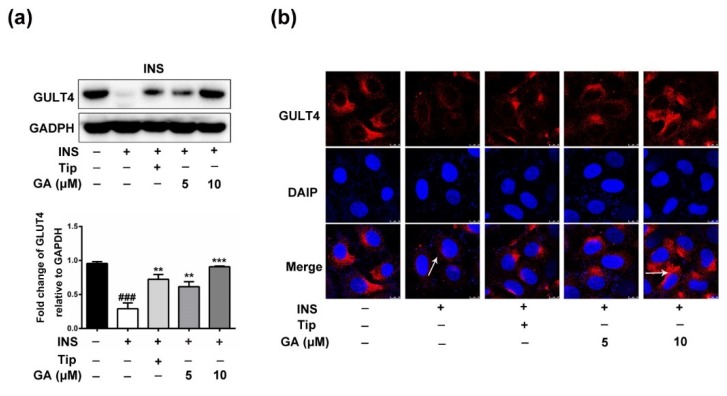
Effect of GA on GLUT4 expression. (**a**) GA significantly promoted the expression of GLUT4 protein. After the cytoplasmic proteins were extracted, the expression level was detected using anti-GLUT4 specific antibodies. (**b**) GA significantly promoted the expression of GLUT4, observed by immunofluorescence staining, and the photographs were captured on an inverted fluorescence microscope. Error bars indicate the mean ± S.D (*n* = 3) (** *p* < 0.01, *** *p* < 0.001, ^###^
*p* < 0.001 compared to the activated insulin signaling pathway group).

**Figure 5 nutrients-11-00604-f005:**
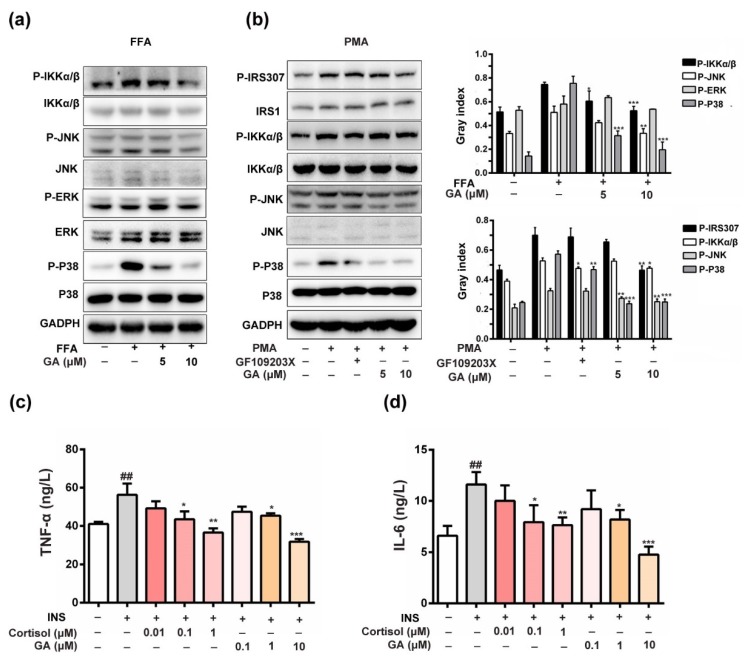
Effect of GA on the anti-inflammatory andinsulin-responsive pathways. (**a**) The total ERK1/2, JNK, p38, and IKK and the phosphorylation of ERK1/2, JNK p38 and IKK were detected in the cells. (**b**) GA blocked the PMA-induced (1 μM) PKC activities and inhibited PKC-induced phosphorylation of IRS1ser307, IKK, p38 and JNK activation. (**c**,**d**) The effect of GA on TNF-α and IL-6 levels in the activated insulin signaling pathway. To examine the effect of GA on the inflammatory cytokines, the levels of TNF-α and IL-6 in the activated insulin signaling pathway was measured by ELISA. HepG2 cells were incubated for 36 h in serum-free DMEM containing 1 µM insulin. The level of TNF-α and IL-6 in the activated insulin signaling pathway was significantly higher compared with the control cells. After treatment with GA (0.1, 1 and 10 µM), the level of TNF-α and IL-6 in the culture medium was significantly lower compared with the activated insulin signaling pathway. Cortisol (0.01, 0.1 and 1 µM) was also able to decrease the levels of TNF-α and IL-6 in the culture medium compared with the activated insulin signaling pathway. The error bars indicate the mean ± S.D (*n* = 3) (* *p* < 0.05, ^##^
*p* < 0.01, ** *p* < 0.01, *** *p* < 0.001 compared to the activated insulin signaling pathway group).

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
