# Peer review of "Glycyrrhetinic Acid Improves Insulin-Response Pathway by Regulating the Balance between the Ras/MAPK and PI3K/Akt Pathways"

_nutrients, 2019, doi:10.3390/nu11030604_

Round 1
Reviewer 1 Report
The authors of this manuscript showed that glycyrrhetinic acid (GA) improve insulin-related signaling and its molecular target signaling. This study is straightforward and the results are relatively clear. However, it is questionable whether the word “insulin resistance” mentioned in the manuscript can be used. Below, the comment about “insulin resistance” and other minor comments are addressed. 1. “Insulin resistance” is the physiological condition in which the insulin-responsive signaling pathway and action is blunted or does not respond at all. Although the stimuli used in the experiment (insulin, TNF-a, and FFA) are clearly reported as substances involved in the pathogenesis of insulin resistance, short-term treatment in the cells with relevant protein levels in vitro should NOT suggest insulin resistance and insulin sensitivity. If the authors want to check for insulin resistance, they should at least be able to conduct in vivo experiments such as GTT, ITT, HOMA-IR, and glucose infusion rate. Considering the in vitro model used by the authors, insulin signaling or insulin-responsive pathway, rather than insulin resistance, will be helpful to convey more accurate information. All the words “insulin resistance” or “insulin sensitivity” used in the direct explanation of the results in this manuscript (except for the description of the reference) must be deleted or modified correctly. For example, - Line 163-164, “As shown in Fig 1, insulin sensitivity increased” - Line 169, legend title of figure 1 - Line 230, “glucose consumption and insulin resistance” - Line 233, “HepG2 cells to establish and insulin resistance model in vitro” - Line 243, “TNF-a-induced insulin resistance model,” - Line 261, legend title of figure 3 - Line276, “on the cell cytoplasm of the insulin resistant cells” - Line295-6, “In this study, FFA-induced insulin resistance,” - Line315, legend title of figure 5 - Line 329, 337-8, 340 - Etc. 2. In materials and method section, there’s no description about the method for separation of cytoplasm and membrane protein (used in Figure 4). The authors should add it. 3. Figure 4A, because the level of the GLUT4 in membrane was not shown, it seems that the total expression level of GLUT4, not the translocation, has changed. Also, in figure 4B, there is only DAPI which is nuclei staining dye. The authors should present membrane specific marker such as E-cadherin to suggest translocation of GLUT4 from cytoplasm to membrane. 4. In figure 2F, 3A, 3C, 3E, I recommend to re-arrange the bar graphs in descending order of concentration (0.1, 1, 10 uM). It would be easier to understand the data by marking them as a consistent trend throughout the manuscript.

Author Response
To reviewer 1:
Thank you very much for your patient review and helpful suggestions. We have carefully considered your suggestions, responded to them point by point below, and revised the manuscript accordingly.
1.“Insulin resistance” is the physiological condition in which the insulin-responsive signaling pathway and action is blunted or does not respond at all. Although the stimuli used in the experiment (insulin, TNF-a, and FFA) are clearly reported as substances involved in the pathogenesis of insulin resistance, short-term treatment in the cells with relevant protein levels in vitro should NOT suggest insulin resistance and insulin sensitivity. If the authors want to check for insulin resistance, they should at least be able to conduct in vivo experiments such as GTT, ITT, HOMA-IR, and glucose infusion rate. Considering the in vitro model used by the authors, insulin signaling or insulin-responsive pathway, rather than insulin resistance, will be helpful to convey more accurate information. All the words “insulin resistance” or “insulin sensitivity” used in the direct explanation of the results in this manuscript (except for the description of the reference) must be deleted or modified correctly. For example,
- Line 163-164, “As shown in Fig 1, insulin sensitivity increased”
- Line 169, legend title of figure 1
- Line 230, “glucose consumption and insulin resistance”
- Line 233, “HepG2 cells to establish and insulin resistance model in vitro”
- Line 243, “TNF-a-induced insulin resistance model,”
- Line 261, legend title of figure 3
- Line276, “on the cell cytoplasm of the insulin resistant cells”
- Line295-6, “In this study, FFA-induced insulin resistance,”
- Line315, legend title of figure 5
- Line 329, 337-8, 340
- Etc.
Response:
Thank you for your professional suggestion. According to your suggestion, we have revised the description and changed the sentence. For example,
① -Line 163-164, We Changed the sentence “As shown in Fig 1, insulin sensitivity increases and glucose production decreases as GA exerts glucocorticoid-like effects.” to " As shown in Fig 1, insulin-responsive pathway improves and glucose production decreases as GA exerts glucocorticoid-like effects."
② -Line 169, legend title of figure 1 , We Changed the sentence “GA affects insulin sensitivity in the whole body.” to " GA affects insulin-responsive pathway in the whole body."
③ -Line 230, We Changed the sentence “Effect of GA on glucose consumption and insulin resistance.” to " Effect of GA on glucose consumption and insulin-responsive pathway."
④ -Line 233, We Changed the sentence “we used insulin-stimulated HepG2 cells to establish an insulin resistance model in vitro.” to " we used insulin-stimulated HepG2 cells to activate the insulin signaling pathway."
⑤ -Line 234, We Changed the sentence “The glucose consumption amount decreased in the case of insulin resistance.” to " The glucose consumption amount decreased when the insulin signaling pathway was activated."
⑥ -Line 245, We Changed the sentence “In Figure 3C, in the TNF-α-induced insulin resistance model.” to " In Figure 3C, in the TNF-α activated the insulin signaling pathway model."
⑦ -Line 261, We Changed the sentence “Effect of GA on glucose consumption and insulin resistance.” to " Effect of GA on glucose consumption and insulin-responsive pathway."
⑧ -Line276, We Changed the sentence “The expression of GLUT4 on the cell cytoplasm of the insulin resistant cells was dramatically reduced.” to " The expression of GLUT4 on the cell cytoplasm of the activated insulin signaling pathway was dramatically reduced."
⑨ -Line295-6, We Changed the sentence “In this study, FFA-induced insulin resistance.” to " In this study, FFA-activated the insulin signaling pathway."
⑩-Line315, We Changed the sentence “Effect of GA on anti-Inflammation and insulin resistance.” to " Effect of GA on anti-Inflammation and insulin-responsive pathway."
⑪ -Line 328, We Changed the sentence “Compared to the insulin resistance group.” to " Compared to the activated insulin signaling pathway group."
⑫ - 337-8, We Changed the sentence “The typical phosphorylation of IRS1Ser307 is thought to mediate insulin resistance.” to " The typical phosphorylation of IRS1Ser307 is thought to mediate insulin signaling pathway."
⑬ -340, We Changed the sentence “such as FFA, TNF-α, and insulin, were used to stimulate HepG2 cells and establish an insulin resistance model..” to " such as FFA, TNF-α, and insulin were used to stimulate HepG2 cells and activated the insulin signaling pathway."
- Etc.
2.In materials and method section, there’s no description about the method for separation of cytoplasm and membrane protein (used in Figure 4). The authors should add it.
Response:
Thank you for your comments. According to your suggestion, I have added a description of the cytoplasmic and membrane protein separation methods in the Materials and Methods section.
Response:
①Membrane and Cytosol Protein Extraction Kit, and BCA Protein Assay Kits were purchased from Beyotime Biotechnology Corporation (Shanghai, Chin-a). (lines 85-86 on page2).
②Protein lysates were obtained according to manufacturer’s suggested protocol of Membrane and Cytosol Protein Extraction Kit, and concentration of proteins were detected
by BCA Protein Assay Kit. (lines 91-93 on page 2).
3. Figure 4A, because the level of the GLUT4 in membrane was not shown, it seems that the total expression level of GLUT4, not the translocation, has changed. Also, in figure 4B, there is only DAPI which is nuclei staining dye. The authors should present membrane specific marker such as E-cadherin to suggest translocation of GLUT4 from cytoplasm to membrane.
Response:
Thank you for your kind suggestions, and we apologize for our Inaccurate description. According your suggestion, we have revised the description and changed the sentence to “To clarify whether GA promote the expression of GLUT4 protein. The HepG2 cells were incubated with 1 μM insulin for 36 h and were then treated with the farnesyltransferase (FTase) inhibitor tipifarnib (Tip, 5 μM) or GA (5, 10 μM) overnight. As shown in Figure 4A, the expression of GLUT4 on the cell cytoplasm of activated insulin signaling pathway was dramatically reduced, which led to a marked reduction in cell glucose uptake. However, compared to the activated insulin signaling pathway group, the Tip (5 μM) and GA (5, 10 μM) groups showed substantially elevated GLUT4 expression on the cell cytoplasm. Furthermore, as demonstrated by immunofluorescence staining, both GA (10 μM) and Tip (5 μM) significantly improved the GLUT4 expression in HepG2 cells (Figure 4B)”. (lines 284-294 on page 8).
4.In figure 2F, 3A, 3C, 3E, I recommend to re-arrange the bar graphs in descending order of concentration (0.1, 1, 10 uM). It would be easier to understand the data by marking them as a consistent trend throughout the manuscript.
Response:
Thank you for your comments. According to your suggestion, The figure have been revised according to the above points.
figure 2F figure 3A
Figure3C figure 3E

Reviewer 2 Report
The objective of the manuscript "Glycyrrhetinic acid improves insulin resistance by regulating the balance between the Ras/MAPK and PI3K/Akt pathways" is sound and the experimental design has sufficient scientific merit. The strength of the manuscript is the findings on potential mechanism of Glycyrrhetinic acid for reversing insulin resistance. The study is well organized and the manuscript presented all relevant results clearly.
There are some minor revisions required to accept this manuscript for publication
1. Need to check the structure of some sentences especially in the first line of sentences and paragraphs (eg. line 159). Better to start with a rationale statement or overall focus of the paragraph.
2. Authors need to clearly describe the data analysis especially why selecting student t test for comparison. The manuscript can be improved with sound statistical model and determination of significance between the means.
3. It would better to have a concluding paragraph with description of significant outcomes, potential limitations and the future direction of the current study.
Overall the manuscript is excellent and will enrich the knowledge on potential health benefits of Glycyrrhetinic acid especially for T2D management.
Author Response
To reviewer #2:
Thank you very much for your patient review and helpful suggestions. We have carefully considered your suggestions, responded to them point by point below, and revised the manuscript accordingly.
1.Need to check the structure of some sentences especially in the first line of sentences and paragraphs (eg. line 159). Better to start with a rationale statement or overall focus of the paragraph.
Response:
Thank you for your comments. According to your suggestion, we have revised the description and changed the sentence to " Emerging data suggest that GA can counteract the development of type 2 diabetes by improving insulin sensitivity. Our results were consistent with the literature, compared with the control, the administration of GA dose-dependently and significantly activated glucocorticoid in mice (Figure 1A)." (lines 166-170 on page 4).
2. Authors need to clearly describe the data analysis especially why selecting student t test for comparison. The manuscript can be improved with sound statistical model and determination of significance between the means.
Response:
Thank you for your kind suggestions, and we apologize for our Inaccurate description. According your suggestion, we have revised the description and changed the sentence to “The results are reported as the means±standard deviation (s.d.). An unpaired, two-tailed Student’s t-test was applied for the statistical comparison of two groups and one-way ANOVA with a Bonferroni correction was used for multiple comparisons. All data were processed using GraphPad Prism statistical software, version 5.01.” (lines 159-162 on page 4).
3. It would better to have a concluding paragraph with description of significant outcomes, potential limitations and the future direction of the current study.
Response:
Thank you for your kind suggestions, I have added concluding paragraphin to “Although natural products have a lot of effects on the prevention and treatment of diabetes, the research on their hypoglycemic mechanism needs further research. Meanwhile, Side effects and toxicity are still a problem to be considered.” (lines 403-405 on page 14).

Round 2
Reviewer 1 Report
I think the authors revised the manuscript accordingly. It includes a more intuitive and scientific discussion of the results than the existing manuscript.